# Fast-spiking GABA circuit dynamics in the auditory cortex predict recovery of sensory processing following peripheral nerve damage

Jennifer Resnik[1,2]*, Daniel B Polley[1,2]*

[1]Eaton-Peabody Laboratories, Massachusetts Eye and Ear Infirmary, Boston, United States; [2]Department of Otolaryngology, Harvard Medical School, Boston, United States

**Abstract** Cortical neurons remap their receptive fields and rescale sensitivity to spared peripheral inputs following sensory nerve damage. To address how these plasticity processes are coordinated over the course of functional recovery, we tracked receptive field reorganization, spontaneous activity, and response gain from individual principal neurons in the adult mouse auditory cortex over a 50-day period surrounding either moderate or massive auditory nerve damage. We related the day-by-day recovery of sound processing to dynamic changes in the strength of intracortical inhibition from parvalbumin-expressing (PV) inhibitory neurons. Whereas the status of brainstem-evoked potentials did not predict the recovery of sensory responses to surviving nerve fibers, homeostatic adjustments in PV-mediated inhibition during the first days following injury could predict the eventual recovery of cortical sound processing weeks later. These findings underscore the potential importance of self-regulated inhibitory dynamics for the restoration of sensory processing in excitatory neurons following peripheral nerve injuries.

*For correspondence:
Jennifer_Resnik@MEEI.HARVARD.
EDU (JR); Daniel_Polley@meei.
harvard.edu (DBP)

**Competing interests:** The authors declare that no competing interests exist.

## Introduction

The enduring plasticity of the adult brain supports a remarkable recovery of perceptual and motor capabilities following peripheral nerve injury. In sensory cortex, reorganization following injury involves a rapid unmasking of inputs from adjacent, undamaged regions of the sensory periphery and, in cases of incomplete injury, a slower, progressive increase in responsiveness to surviving nerve fibers within the damaged region (*Rasmusson, 1982*; *Merzenich et al., 1983*; *Gilbert and Wiesel, 1992*; *Kaas et al., 1990*; *Robertson and Irvine, 1989*; *Qiu et al., 2000*; *Chambers et al., 2016*; *Calford and Tweedale, 1988*). The initial stages of this reorganization may be enabled by homeostatic processes that compensate for dramatic swings in afferent input to maintain neural excitability around a set point (*Nahmani and Turrigiano, 2014*). Recordings from cortical pyramidal neurons in culture or acute brain slices following sudden shifts in excitatory input have revealed a coordinated sequence of changes at glutamatergic and GABAergic synapses that normalize firing rates and rebalance network activity (*Kilman et al., 2002*; *O'Brien et al., 1998*; *Turrigiano et al., 1998*; *Xue et al., 2014*; *D'amour and Froemke, 2015*; *Li et al., 2014*). Whether and how synaptic and extra-synaptic modifications underlie reorganized sensory processing in deafferented zones of the adult cortex remains to be determined, but accumulating evidence from intact preparations suggests that the stabilization of new synaptic inputs from spared regions of the sensory periphery may emerge through a combination of rapid disinhibition and increased structural motility (*Li et al.,*

*2014*; *Marik et al., 2014*; *Yamahachi et al., 2009*; *Darian-Smith and Gilbert, 1994*; *Keck et al., 2011*, *2013*; *Garraghty et al., 1991*; *Florence et al., 1998*).

In the auditory system, a loss of cochlear hair cells and/or cochlear nerve afferent synapses is associated with reduced GABA signaling and cortical hyperexcitability, as manifest in elevated spontaneous firing rates, enhanced central gain and unmasking of sound-evoked responses from undamaged regions of the cochlea (*Qiu et al., 2000*; *Chambers et al., 2016*; *Seki and Eggermont, 2003*; *Yang et al., 2012*; *Scholl and Wehr, 2008*). However, these biomarkers have not been compared with one another over the full course of functional recovery with fine-grain temporal resolution at the level of individual neurons. As a result, it is unclear whether dynamic changes in intracortical inhibition precede, follow, outlast or recede ahead of functional changes in receptive field mapping and neural hyperexcitability. Nor is it clear whether the decline in GABA markers following peripheral injury arise from a reduced influence of PV-expressing fast-spiking interneurons, which synapse onto pyramidal neuron somata and have a well-recognized role in gating activity-dependent cortical plasticity, or instead from other GABA neuron subtypes, which can synapse onto other GABA neurons and exert a net excitatory effect on network activity (*Isaacson and Scanziani, 2011*). Here, we address these unanswered questions by measuring changes in PV-mediated intracortical inhibition alongside hyperexcitability and receptive field plasticity from individual regular spiking (RS) putative pyramidal neurons over a several month period surrounding varying degrees of auditory nerve damage. These findings highlight a rapid loss and recovery in PV-mediated inhibition that may compensate for a sudden drop in afferent drive following cochlear afferent loss to support a progressive recovery of sensory processing from spared nerve fibers.

## Results

We implanted optetrode assemblies into the primary auditory cortex (A1) of adult PV-Cre:Ai32 mice (*Figure 1a*) to isolate regular spiking (RS) putative pyramidal neurons based on their spike shape and estimate the strength of local inhibition on RS units during optogenetic activation of PV–expressing GABA neurons (*Figure 1b–c*). The mechanical stability of the optetrode assembly combined with the thin, flexible tetrode wires allowed us to isolate single units (*Figure 1d*) and hold them for many weeks, according to a conservative, objective statistical standard (*Figure 1e–f*). With this approach, we could measure auditory responsiveness and local inhibitory tone from individual RS neurons in awake, head-fixed adult mice over a 7–8 week period surrounding varying degrees of auditory nerve injury (*Figure 1g*). Although we could occasionally isolate PV-expressing FS interneurons on our tetrodes (*Figure 1b–c*), we could rarely hold these neurons for more than a single recording session (refer to *Figure 1—figure supplement 1*). Therefore, all descriptions of long-term changes in single unit response properties were limited to RS, putative pyramidal neurons.

Most forms of cochlear injury induce an intractable set of changes to the auditory nerve as well as sensory and non-sensory cells within the cochlea. In cases of widespread cochlear damage, it is impossible to attribute abnormal cortical tuning to a central plasticity versus abnormal cochlear filtering or amplification. We circumvented this problem by selectively eliminating afferent nerve fibers without permanently affecting cochlear mechanics using a well-characterized noise exposure protocol that lesions approximately 50% of high-frequency afferent nerve synapses in the 16–45 kHz region of the cochlea without damaging hair cells (n = 4, *Figure 2a*) (*Wan et al., 2014*; *Kujawa and Liberman, 2009*). This protocol induces only a temporary elevation of the auditory brainstem response (ABR) and distortion product otoacoustic emission (DPOAE) thresholds, but a permanent reduction in ABR wave 1 amplitude, which has been shown to reflect the permanent elimination of high-frequency cochlear afferent synapses (*Figure 2b*, the reader is referred to the figure legends and *Supplementary file 1* for most statistical reporting) (*Liberman and Kujawa, 2017*).

We implanted optetrode assemblies into the 32 kHz region of the A1 tonotopic map and recorded RS units with high-frequency receptive fields, low-intensity thresholds to broadband noise stimulation and strong feedforward inhibition from PV neurons (*Figure 2c–e*, top row). Several hours after noise exposure that affects high-frequency regions of the cochlea, the same unit exhibited elevated response thresholds for a broadband noise burst, slightly reduced PV-mediated inhibition and the appearance of a new, well-tuned low-frequency receptive field that was absent just hours earlier (*Figure 2c–e*, second row). We quantified the invasion of spared, low-frequency inputs and the eventual restoration of the original receptive field by measuring daily changes in frequency response area

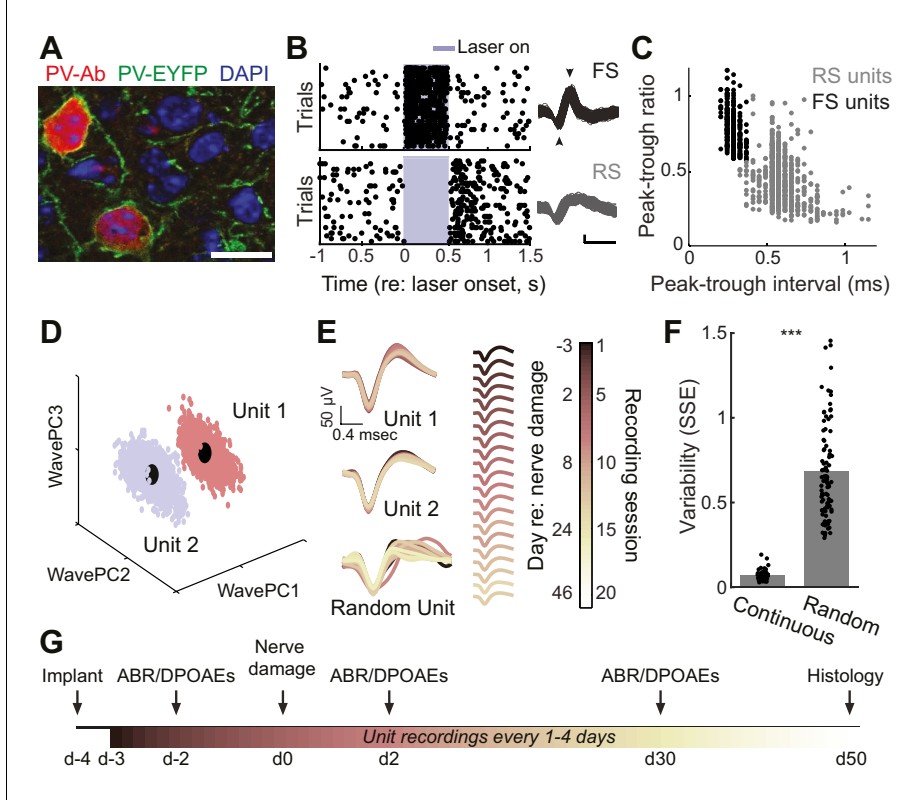

**Figure 1.** Approach for chronic single unit recordings and optogenetic activation in mouse A1. (**A**) Immunolabeling of PV neurons in A1 with co-localization of EYFP reporter in Pv-Cre:Ai32 transgenic mice. DAPI labels cell nuclei. Scale bar = 15 μm. (**B**) Spike raster plots illustrating that optogenetic activation of fast spiking (FS) PV+ units (black, top) inhibits regular spiking (RS) units (gray, bottom). *Right,* spike waveforms for the RS and FS units. Arrowheads denote spike peak and trough. Scale bars, 0.5 ms and 50 μV. (**C**) Scatter plot showing the bimodal distribution of peak-tough amplitude and timing differences across all RS (gray) and FS (black) units. (**D**) A random sub-sampling of spike waveforms recorded over 53 days from a single wire of a tetrode projected down into the first three principal components (PC). (**E**) Spike waveforms from the two units identified in (**D**) across all recording sessions, color-coded and superimposed chronologically. Waveforms for random units were selected at random from all simultaneously recorded units. (**F**) The variability in the actual unit waveforms, estimated as the sum of squared errors (SSE), is significantly less than randomly shuffled units (p<0.001, f(1)=814.73, mean ± SEM). (**G**) Experimental design.

The following figure supplement is available for figure 1:

**Figure supplement 1.** Long-term tetrode recordings from isolated single units is feasible with RS neurons, but not FS neurons.

---

(FRA) overlap with the baseline FRA as well as tone-evoked firing rates from the spared (4–6 kHz) and partially denervated (32–48 kHz) cochlear regions (*Figure 2f*, left). We compared these metrics of receptive field reorganization against the typical biomarkers of hyperexcitability in the auditory pathway: decreased response thresholds, increased spontaneous firing rate and increased gain, defined here as the slope of the stimulus input-output function (*Figure 2f*, middle). Finally, we related these markers of RS unit hyperexcitability and receptive field plasticity to changes in the strength of PV-mediated inhibition (*Figure 2f*, right; n = 208).

Several hours after noise exposure, firing rates evoked by stimulation of spared, low-frequency cochlear regions had nearly doubled (increased by 96.7%, p=0.04) with a small, but significant decrease in inhibition (−9.8%, p=0.02) and no significant change in gain or spontaneous rate (*Figure 2g*). Six to ten days after noise exposure, inhibition had decreased by 40% (p<0.0001), receptive fields remained focused on spared regions of the cochlea (p<0.01) and spontaneous firing

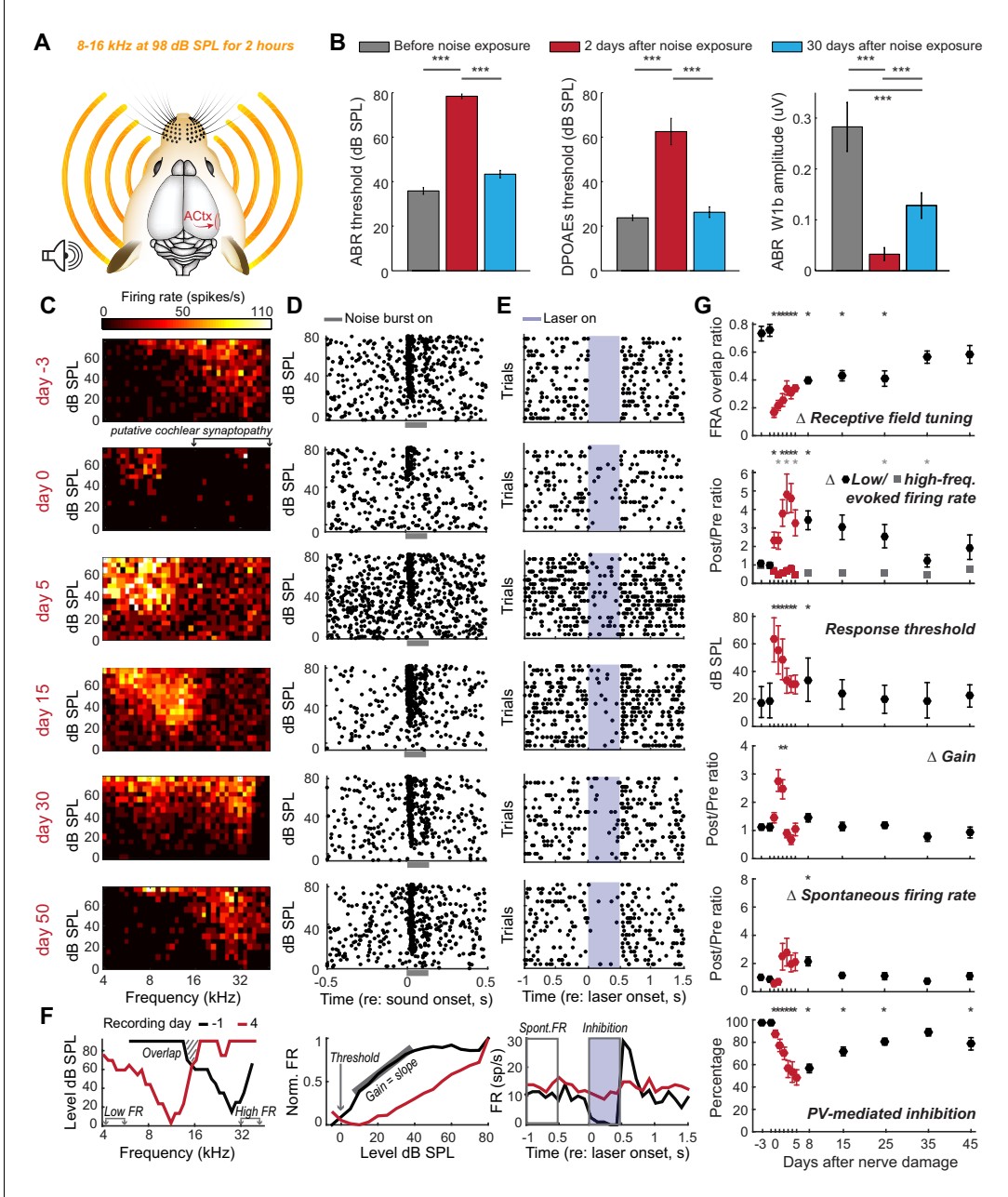

**Figure 2.** Moderate loss of high-frequency auditory nerve fibers induces striking – but partially reversible - receptive field reorganization and inhibition. (A) Adult mice (16 weeks) were exposed to octave-wide noise at 98 dB SPL for 2 hr, which has been shown to eliminate approximately 50% of auditory nerve synapses onto inner hair cells in the 16–45 kHz region of the cochlea (*Wan et al., 2014*; *Kujawa and Liberman, 2009*). (B) Mean ± SEM ABR (left) and DPOAE thresholds (middle) were temporarily elevated 2 days after noise exposure but recover fully by 30 days. ABR wave 1b amplitude reflects the permanent loss of Type-I spiral ganglion nerve fibers (right). (C) A frequency response area (FRA) from a single unit recorded in the high-frequency zone of the A1 tonotopic map rapidly assumed a low-frequency FRA hours after noise exposure but shifted back towards baseline tuning over the ensuing month. (D–E) Rasters depict spiking from the same RS unit evoked by broadband noise at varying sound levels (D) or inhibited by optogenetic activation of neighboring PV+ neurons (E). Note temporary threshold shift and change in inhibition strength. (F) Quantification approach illustrated from a different single unit recorded before (black) and after (red) noise exposure. (G) Mean ± SEM (n = 208 units) values for each response property illustrated in (F). Data for each unit are normalized to the mean value measured during the 3-day period prior to noise exposure. Red symbols are used to visually highlight the rapid plasticity occurring during the first 5 days following nerve damage. Asterisks = post-hoc pairwise comparison between identified groups (B) or to baseline value (G) p<0.05 after correcting for multiple comparisons.

rate was significantly elevated (p<0.05). Central gain, spontaneous firing rate and frequency tuning all shifted back towards baseline values as response thresholds in the cochlea, ABR and cortical neurons recovered to pre-exposure levels. By the time of the final recording session, approximately 7 weeks after noise exposure, reduced PV-mediated inhibition (−19%, p<0.001) was the only enduring compensatory plasticity marker for the putative loss of high-frequency auditory nerve synapses.

Exposure to ototoxic drugs can cause a sudden and extreme loss of afferent signaling from the cochlea. Would homeostatic adjustments support as complete a recovery of function following a massive loss of auditory nerve fibers? We addressed this question by applying ouabain bilaterally to the cochlear round window membrane (*Figure 3a*). In mice, ouabain selectively and evenly eliminates >95% of Type-I spiral ganglion afferent neurons across the frequency map, without affecting other types of sensory and non-sensory cells in the cochlea or auditory nerve (*Yuan et al., 2014*). Bilateral ouabain treatment virtually eliminated the ABR without any adverse DPOAE effects in all mice (n = 6, *Figure 3b*), but, interestingly, was associated with a slow, partial recovery of sound-evoked cortical spiking in half the mice (*Figure 3c*) but an unremitting loss of auditory responsiveness in the other half (*Figure 3d*).

When DPOAE thresholds are normal, ABR wave 1b amplitude is an accurate proxy for the number of cochlear afferent synapses (*Wan et al., 2014*; *Kujawa and Liberman, 2009*; *Liberman and Kujawa, 2017*). However, the ABR wave 1b amplitude was not significantly different between mice that recovered auditory processing following ouabain treatment versus those that did not (unpaired t-test, p>0.4; see also *Figure 3—figure supplement 1*). Whereas our estimate of peripheral denervation bore no relationship to the mode of cortical recovery, early dynamics in inhibitory tone and spontaneous firing rate were linked to the recovery of cortical processing. RS units that eventually recovered auditory thresholds exhibited a transient spike in spontaneous firing rate, a sustained increase in central gain and a steep drop in PV-mediated inhibition that began to recover as high-threshold sound-evoked activity returned (*Figure 3e–h*, top row, n = 156). By contrast, in RS units that never recovered auditory sensitivity, changes in spontaneous rate were comparatively sluggish, while inhibitory strength declined gradually and monotonically (*Figure 3e–h*, middle row, n = 169). This stark dichotomy in the cortical response to nerve injury was shared among all units recorded from a given mouse; if one unit recovered auditory responsiveness, all units recovered auditory responsiveness (*Figure 4*). As a negative control in a separate group of mice (n = 3), we applied sterile water to the round window membrane rather than ouabain and confirmed that cortical response properties were stable over the recording period in mice with intact auditory nerves (*Figure 3e–h*, bottom row, n = 156).

These findings highlight the coordination of short- and long-term changes in adult A1 following peripheral nerve injury. Spontaneous rate, central gain and inhibition all scaled with the degree of nerve damage (*Figure 3f–h* versus *Figure 2g*, *Figure 4*), but varied substantially both over time and between mice. Restored auditory sensitivity was associated with a sharp loss and partial recovery of inhibition from PV interneurons, where the degree of eventual recovery could be predicted from changes in inhibition ~1 week after nerve damage (*Figure 4a*). Not only was PV-mediated inhibition a predictor of sensory recovery for both mild and severe peripheral damage, but also it was a better predictor than were modifications in spontaneous activity (*Figure 4b*) or central gain, the biomarkers most commonly employed to assess central auditory plasticity following sensorineural hearing loss (refer to *Figure 4—figure supplement 1* for an expanded analysis of predictors of recovery).

## Discussion

In this study, we tracked a plasticity in PV-mediated inhibition and single neuron responsiveness in the primary auditory cortex of adult mice following a sudden loss of input from the auditory nerve. Just hours after noise exposure, we observed the appearance of new, low-threshold receptive fields tuned to undamaged regions of the cochlea, accompanied by a small, significant drop in PV-mediated inhibition. Over the ensuing weeks, as cochlear thresholds and cortical receptive field tuning returned to baseline conditions, we noted only a brief spike in traditional biomarkers of central auditory hyperactivity, such as increased spontaneous firing rate and increased central gain. By contrast, PV-mediated inhibition remained reduced through our final recording session and was the sole physiological indicator for the permanent, presumed loss of high-frequency cochlear synapses. Importantly, this is not to say that auditory response normalization was mediated by disinhibition; our

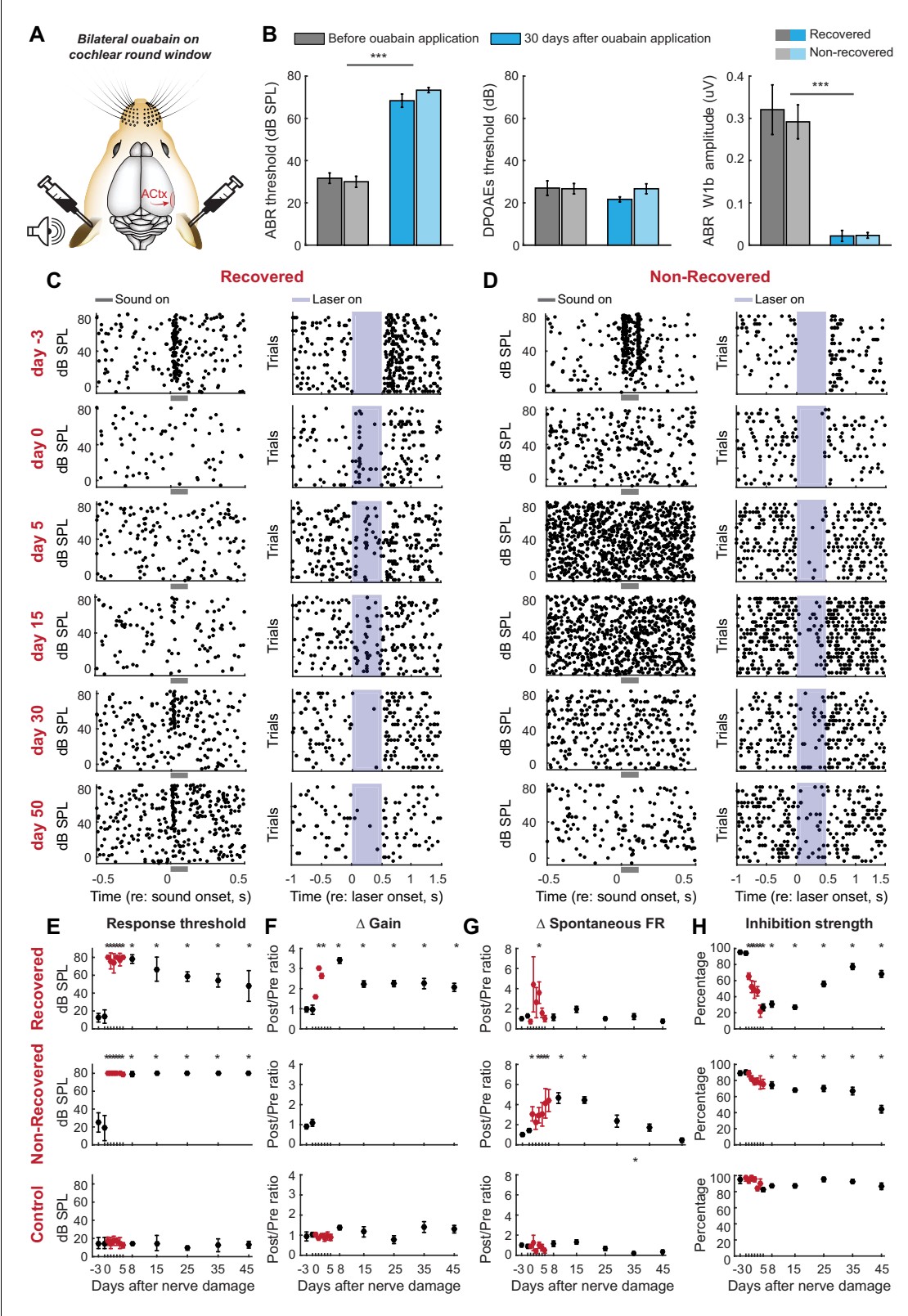

**Figure 3.** Gradual and variable recovery of auditory processing following a massive, bilateral loss of cochlear afferent nerve fibers can be predicted from early changes in inhibitory strength. Plotting conventions follow *Figure 2*. (A) Ouabain was applied bilaterally to the round window membrane. (B) Mean ± SEM ABR thresholds were substantially elevated 30 days after ouabain application (*left*) but DPOAE thresholds were unaffected, indicating normal outer hair cell function (*middle*). ABR wave1b was virtually eliminated after ouabain (*right*). Darker and lighter shading represent measurements

*Figure 3 continued on next page*

*Figure 3 continued*

from mice that recovered cortical sound thresholds versus mice that did not, respectively. (**C–D**) Rasters document changes in noise-evoked spiking and laser-induced inhibition, respectively, from single A1 RS units recorded over a 53-day period from a mouse that eventually recovered function (**C**) and a mouse that did not recover function (**D**). (**E–H**) Mean ± SEM noise-evoked thresholds (**E**), rate-level function gain (**F**), spontaneous firing rate (**G**) and inhibition strength (**H**) for all units recorded from three mice that recovered (top row, n = 156), three mice that did not recover (middle row, n = 169) or three mice that underwent a sham surgery (bottom row, n = 156). Asterisks = post-hoc pairwise comparison between identified groups (**B**) or to baseline value (**E–H**) p<0.05 after correcting for multiple comparisons.

The following figure supplement is available for figure 3:

**Figure supplement 1.** ABR wave 1 amplitude scales with the degree of auditory nerve damage, but does not vary systematically between mice that recover cortical sound thresholds versus those that do not.

observations are correlative and do not speak to the necessity nor sufficiency of changes in PV networks for functional recovery. Compensation for peripheral damage could have arisen from post-synaptic changes in GABA receptors in the RS principal neurons recorded here (*Sarro et al., 2008*) or from presynaptic changes in the GABA neurons themselves (*Marik et al., 2014*; *Keck et al., 2011*; *Hengen et al., 2013*), as well as many other homeostatic plasticity mechanisms in the RS neurons including upward scaling of glutamatergic synapses (*O'Brien et al., 1998*; *Turrigiano et al., 1998*), changes in extrasynaptic GABA or glutamate receptors that mediate tonic currents (*Chen et al.,*

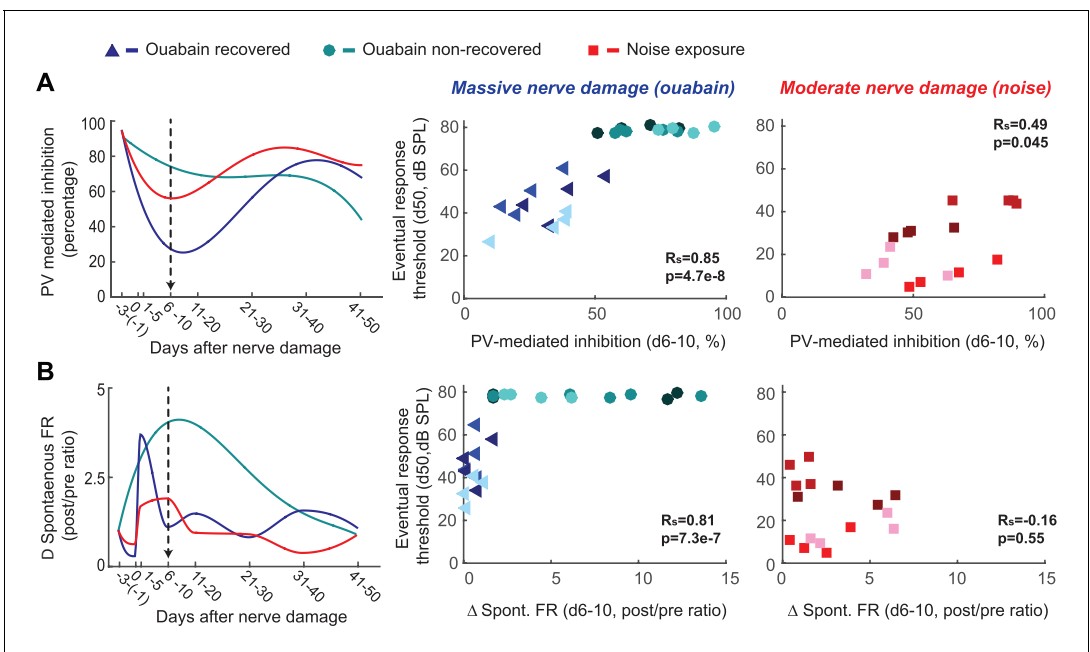

**Figure 4.** Early changes in PV-mediated inhibitory strength predict eventual recovery of cortical sound processing after nerve damage. (**A–B**) PV-mediated inhibition strength (**A**) and change in spontaneous firing rate (**B**) timelines for sound exposed, ouabain recovered and ouabain non-recovered mice (red, blue and teal accordingly, plotted for visualization purposes as a fourth order polynomial fit to the data presented *Figures 2* and *3*). (*Middle and Right*) Correlation between broadband noise threshold at day 50 and PV-mediated inhibition (**A**) or spontaneous firing rate change (**B**) at days 6–10 for ouabain-treated (middle) and sound-exposed mice (right, Spearman's correlation coefficient, Rs). Each symbol represents a single RS unit. The shape and color correspond to a particular group, as described above. The shading corresponds to a particular mouse within that group.

The following figure supplement is available for figure 4:

**Figure supplement 1.** Auditory cortex units show robust PV-mediated inhibition and low-threshold sensory responses prior to nerve damage, both of which are unrelated to the eventual response thresholds after nerve damage.

2010; *Fleming et al., 2011*; *Sametsky et al., 2015*), or changes in intrinsic membrane response properties (*Turrigiano et al., 1994*; *Karmarkar and Buonomano, 2006*; *Li et al., 2015*).

In classic and contemporary studies of adult cortical plasticity following deafferentation, fine-scale reorganization varied widely between local subnetworks of RS neurons and the completeness of global remapping was negatively correlated with the spatial extent of peripheral lesions (*Robertson and Irvine, 1989*; *Merzenich et al., 1984*; *Barnes et al., 2015*; *Keck et al., 2008*). Here, we report that the mode of cortical sensory recovery (complete, partial or none) was roughly shared between all RS units recorded in a given mouse and did not strictly depend on the estimated auditory nerve damage. Unlike focal ablations of the retina, basilar membrane or skin surface, cochlear ouabain treatment eliminates 95% of primary afferent synapses without affecting sensory transduction mechanisms (*Chambers et al., 2016*; *Yuan et al., 2014*). This provided us with an uncommon opportunity to track the restoration of sensory responses transmitted through the small fraction of surviving nerve fibers, rather than the typical approach of describing competitive reorganization from ectopic, spared inputs neighboring the lesion. As compared to the rapid unmasking of excitatory inputs from neighboring undamaged regions of the sensory organ, which occurs within hours (*Figure 2*, see also [*Gilbert and Wiesel, 1992*; *Calford and Tweedale, 1988*]), the recovery of function for spared afferent inputs interspersed within a damaged portion of the nerve, if it occurred at all, unfolded over several weeks. The variable modes of recovery between mice with carefully matched nerve damage may relate to earlier observations that cortical sensory reorganization is not only determined by the pattern and density of peripheral innervation, but may also be directed by activity-dependent differences that are orchestrated through behavioral use (*Clark et al., 1988*; *Xerri et al., 1998*; *Polley et al., 1999*).

Following either type of nerve damage, the cortical networks that recovered function were those that matched the sudden drop in bottom-up excitatory drive with a rapid dip in PV-mediated inhibition. Precisely balanced excitation and inhibition enables a wider dynamic range of sensory information coding (*Isaacson and Scanziani, 2011*; *Shadlen and Newsome, 1998*; *Pouille et al., 2009*; *Zhou et al., 2014*) and supports adaptive plasticity without forsaking network stability (*Vogels et al., 2011*; *Froemke et al., 2013*; *Hellyer et al., 2016*; *Barrett et al., 2016*). These findings suggest that self-regulating PV circuits may not only play an important role in rebalancing network activity in response to reduced afferent drive during developmental critical periods (*Xue et al., 2014*; *Yazaki-Sugiyama et al., 2009*), but in the adult cortex following sensory nerve damage as well.

## Materials and methods

### Animals and cochlear denervation
All procedures were approved by the Animal Care and Use Committee at the Massachusetts Eye and Ear Infirmary and followed guidelines established by the National Institutes of Health for the care and use of laboratory animals. Subjects included 13 PV-Cre:Ai32 mice of either sex (a cross between B6;129P2-*Pval*btm1(cre)Arbr/J and Ai32 (RCL-ChR2(H134R)/EYFP), Jackson Laboratory), aged 16 weeks at the time of optetrode implantation.

### Bilateral auditory nerve damage
#### Acoustic exposure
The acoustic over-exposure stimulus was an octave band of noise (8–16 kHz) presented at 98 dB sound pressure level (SPL) for 2 hr. During exposures, animals were awake and unrestrained within a 12 × 10 cm, acoustically transparent cage. The cage was suspended directly below the horn of the sound-delivery loudspeaker in a small, reverberant chamber. Noise calibration to target SPL was performed immediately before each exposure session.

#### Ouabain
Selective elimination of Type-I spiral ganglion neurons was achieved by applying a 1 mM solution of ouabain octahydrate (Sigma) and sterile water to the left and right round window niche, as described previously (*Yuan et al., 2014*). Animals were anesthetized with ketamine (120 mg/kg) and xylazine (12 mg/kg), with half the initial ketamine dose given as a booster when required. The connective

tissue and underlying muscle were blunt dissected and held away from the bulla with retractors. A small opening was made in the bulla with the tip of a 28.5-gauge needle. The exposed round window niche was filled with 1–2 μL of the ouabain solution using a blunted needle. Ouabain was reapplied five more times at 15 min intervals, wicking the remaining solution away with absorbent paper points before each application. For control experiments, sterile water was placed on the cochlear round window using an identical procedure. Measurements of the auditory brainstem response (ABR) and distortion product otoacoustic emission (DPOAE, see subsequent Experimental Procedures) were made during the ouabain application to confirm functionality and ABR threshold shift without changes in DPOAE thresholds or amplitudes. Additional ouabain was applied, as necessary, until the ABR threshold at 16 kHz was greater than 55 dB sound pressure level (SPL). The incision was sutured and the mouse was given an analgesic (Buprenex, 0.05 mg/kg) before being transferred to a heated recovery chamber.

## Cochlear function tests

Mice were anesthetized with ketamine and xylazine (as above), and placed on a homeothermic heating blanket during testing. ABR stimuli were 5 ms tone pips (8, 16 and 32 kHz pure tones) with a 0.5 ms rise-fall time delivered at 30 Hz. Intensity was incremented in 5 dB steps, from 20–80 dB SPL. ABR threshold was defined as the lowest stimulus level at which a repeatable waveform could be identified. DPOAEs were measured in the ear canal using primary tones with a frequency ratio of 1.2, with the level of the f2 primary set to be 10 dB less than f1 level, incremented together in 5 dB steps. The 2f1-f2 DPOAE amplitude and surrounding noise floor were extracted. DPOAE threshold was defined as the lowest of at least two continuous $f_2$ levels, for which the DPOAE amplitude was at least two standard deviations greater than the noise floor. All treated animals underwent rounds of DPOAE and ABR testing before, 2 days and approximately 30 days after the procedure.

## Immunohistochemistry

Mice were perfused with 0.01 M phosphate buffered saline (PBS) (pH = 7.4) followed by 4% paraformaldehyde in 0.01 M PBS. Brains were removed and stored in 4% paraformaldehyde for 12 hr before transferring to cryoprotectant (30% sucrose) for 48 hr. Sections (40 μm) were cut using a cryostat (Leica CM3050S). Sections were washed in PBS containing 0.1% Triton X-100 (three washes, 5 min each), incubated at room temperature in blocking solution (Super Block) for 6 min at room temperature and then incubated in primary antibody (PV 27 rabbit anti-parvalbumin, Swant,1:1000 dilution) at four degrees C overnight. The next day, slices were washed and incubated in secondary antibody (Alexa 647 goat anti-rabbit immunoglobulin G, Invitrogen, 1:200 dilution) for 1.5 hr at room temperature. Sections were mounted on gelatin-subbed glass slides (BBC Biochemical) and coverslipped. Fluorescence images were obtained with a confocal microscope (Leica).

## Chronic optetrode implantation

Mice were brought to a surgical plane of anesthesia with ketamine/xylazine, as described above. Body temperature was maintained at 36.5°C with a homeothermic blanket system. Using a scalpel, a small craniotomy was centered over the right auditory cortex leaving the dura mater intact. The brain surface was covered with sterile ointment. Chronic implants consisted of a modified microdrive array (VersaDrive 4, Neuralynx Inc) containing 4 independently moveable, closely spaced tetrodes and an optic fiber (0.2 mm, Thorlabs, Inc.). The optic fiber was mounted into the microdrive assembly so that it rested several millimeters above brain the surface. Tetrodes were four twisted nichrome wires (12.5 μm in diameter, Stablohm 650 wire, California Fine Wire Company), electroplated with gold to reach an impedance of 0.5–1 MΩ (nanoZ and ADPT-NZ-VERSA adapter, Neuralynx Inc). Microdrive arrays were positioned over the right primary auditory cortex based on an initial mapping of tonotopic gradient orientation using independently moveable tungsten microelectrodes (*Hackett et al., 2011*). Tetrodes were lowered into the middle cortical layers to match the approximate depth of tungsten microelectrodes recordings. Silver wire was fixed in place atop the left frontal cortex to serve as a ground. A titanium head plate was affixed to the skull once the microdrive assembly was firmly in place with dental cement (C&B Metabond).

## Neurophysiology data collection

Single unit recordings were made from awake, head-fixed mice. Mice were continuously video monitored during recording. Raw neural signals were digitized at 32-bit, 24.4 kHz (RZ5 BioAmp Processor; Tucker-Davis Technologies) and stored in binary format for offline analysis. The signal was bandpass filtered at 300–3000 Hz with a second-order Butterworth filter and movement artifacts were minimized through common mode rejection. Spike waveforms were extracted based on a threshold and sorted offline into single units using Wave_clus, a semiautomatic clustering algorithm (*Quiroga et al., 2004*). Sorts were performed on concatenated data files from all recording sessions. Neurons were considered continuously recorded if spike sorting from data across the entire experiment yielded a single unit by the following criteria: (1) Waveforms constituted a statistically isolated cluster when all waveforms from all recording sessions were considered; (2) Biophysical properties consistent with single units, such as an absolute refractory period, were met; (3) High uniformity in spike shape was maintained, as assessed by a comparison of the sum of squared errors (SSE) across days (fractional difference in amplitude squared); and (4) Consistently high signal to noise ratios.

The first recordings after auditory nerve injury were made approximately 2 hr after the end of noise exposure and approximately 14 hr after ouabain application (to allow for extra recovery from injectable anesthetics). Recordings were made every 1–4 days thereafter. We only included units that were active in at least 75% of recording sessions and whose firing rate was stable for the two baseline recording sessions prior to peripheral damage. After offline sorting the mean waveform of neurons was calculated, the tetrode wire on which the amplitude of the spikes was largest was used. The trough-to-peak interval and the peak trough ratio were calculated and units were classified as fast-spiking PV+ units or RS-putative excitatory units. The classification was confirmed by laser response in the pre-condition. All subsequent analyses were performed in MATLAB 2015a (MathWorks).

## Acoustic stimuli

Stimuli were generated with a 24-bit digital-to-analog converter (National Instruments model PXI-4461). For DPOAE and ABR tests, as well as during electrode implant surgery, stimuli were presented via in-ear acoustic assemblies consisting of two miniature dynamic earphones (CUI CDMG15008–03A) and an electret condenser microphone (Knowles FG-23339-PO7) coupled to a probe tube. Stimuli were calibrated in the ear canal in each mouse before recording. In awake recordings, stimuli were presented via a free-field electrostatic speakers (Tucker-Davis Technologies) facing the left (contralateral) ear. Free-field stimuli were calibrated before recording with a wideband ultrasonic acoustic sensor (Knowles Acoustics, model SPM0204UD5).

## Neurophysiology data analysis

### Rate-level functions

Broadband noise tokens (4–64 kHz, 0.1 s duration, 4 ms raised cosine onset/offset ramps) were presented at 0–80 dB SPL in 5 dB increments. Threshold was defined as the lowest of at least three continuous stimulus levels for which the response to sound was significantly higher than the spontaneous activity. Gain was defined as the relationship between sound level (input) and firing rate (output). The gain was measured as the slope of the linear fit of the initial rising phase from the rate-level function.

### Frequency response areas

FRAs were delineated using pseudorandomly presented tone pips (50 ms duration, 4 ms raised cosine onset/offset ramps, 0.5–1 s intertrial interval) of variable frequency (4–48 kHz in 0.1 octave increments) and level (0–75 dB SPL in 5 dB increments). Each tone pip was repeated three times and responses to each iteration were averaged. The start of the spike collection window was set to be the point when the firing rate began to consistently exceed the spontaneous rate by at least 3 SD. The offset of the spike collection window was the first bin after the response decreased to less than 4 SD above the spontaneous rate. Additional details on spike windowing and FRA boundary determination are described elsewhere (*Guo et al., 2012*). Receptive field overlap was defined as the number of frequency–intensity combinations in the intersection of the reference and comparator FRA divided by the number of points contained within their union. The reference FRA was the

average of the pre-damage FRAs for a given unit and the comparator was selected from each individual measurement time.

## Optogenetic activation

Collimated blue light (473 nm, 500 ms duration, 12 mW) was generated by a laser (DPSS, LaserGlow Co.) and delivered to the brain surface via an optic fiber coupled to the tetrode assembly. Precise timing of the light pulse was controlled via a custom shutter system. The intensity of the blue laser was calibrated for each implant prior to the start of recording with a photodetector (Thorlabs, Inc.). Percentage of inhibition in excitatory cells was calculated as the percentage of 20 ms bins, during the 500 ms laser duration, that were significantly suppressed when compared to spontaneous firing rate (paired t-test corrected for multiple comparisons).

## Statistical analysis

A one-way ANOVA or mixed design ANOVA was employed for determining statistical significance (Matlab). Post-hoc pairwise comparisons were corrected for multiple comparisons using the Bonferroni correction. All error bars are mean ± SEM. Pearson and Spearman correlation coefficients were used for correlations of normally and not normally distributed variables accordingly.

## Acknowledgements

We thank Prof. Nao Uchida for assistance with optetrode fabrication and Dr. Ken Hancock for support with data collection software. This work was supported by an EMBO postdoctoral fellowship (JR), R01 DC009836 (DBP), a research grant for Autifony Therapeutics (DBP) and the Lauer Tinnitus Research Center (DBP).

## Additional information

### Funding

| Funder | Grant reference number | Author |
| --- | --- | --- |
| European Molecular Biology Organization | Long term postdoctoral fellowship | Jennifer Resnik |
| National Institute on Deafness and Other Communication Disorders | RO1 DC009836 | Daniel B Polley |

The funders had no role in study design, data collection and interpretation, or the decision to submit the work for publication.

### Author contributions

JR, Conceptualization, Data curation, Formal analysis, Funding acquisition, Methodology, Writing—original draft, Writing—review and editing; DBP, Conceptualization, Funding acquisition, Methodology, Writing—original draft, Project administration, Writing—review and editing

### Author ORCIDs

Jennifer Resnik, http://orcid.org/0000-0002-0573-0008
Daniel B Polley, http://orcid.org/0000-0002-5120-2409

### Ethics

Animal experimentation: All procedures were approved by the Animal Care and Use Committee at the Massachusetts Eye and Ear Infirmary (protocol number 10-03-006) and followed guidelines established by the National Institutes of Health for the care and use of laboratory animals. All surgeries were performed under ketamine and xylazine, and every effort was made to minimize suffering.

## Additional files

**Supplementary files**
• Supplementary file 1. Statistical reporting. Detailed description of statistical analysis for all figures and figure supplements.

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
