## [Decision Letter]

Thank you for submitting your article "Coordinated homeostatic modifications enable cortical compensation for sensory nerve damage" for consideration by *eLife*. Your article has been favorably evaluated by Andrew King (Senior Editor) and three reviewers, one of whom is a member of our Board of Reviewing Editors. The reviewers have opted to remain anonymous.

The reviewers have discussed the reviews with one another and the Reviewing Editor has drafted this decision to help you prepare a revised submission.

Summary:

This manuscript describes an analysis of the behavior of regular spiking (RS) neurons in primary auditory cortex following two forms of peripheral injury to the cochlea. In a recent report from the same group (Chambers et al., 2016), they showed that the central auditory system can compensate for loss of sensory input, enabling cortical neurons to respond relatively normally to sounds after ~95% of Type I afferents were destroyed by applying ouabain to the round window. Here they examined the time course of changes in RS cell spontaneous activity, responsiveness and degree of inhibition by PV interneurons (using ChR2-mediated stimulation in mice engineered to express ChR2 only in PV interneurons: PV-Cre x Ai32 mice) following two forms of cochlear injury. They find that PV-mediated inhibition of RS cells is transiently reduced following noise-induced trauma, which recovers along with normal response properties of the neurons. In response to bilateral ouabain application, which induces more extensive damage to afferents, they find that some cells were able to partially recover responsiveness to sound. These cells exhibited a greater initial reduction in PV mediated inhibition (75-80% versus 30% inhibition) and correspondingly higher rates of spontaneous activity than RS cells that did not recover, suggesting that homeostatic changes in the strength of PV interneuron-mediated inhibition controls the recovery from these injuries. The design and analyses are thoughtful and well-executed, and the results are of importance to our general understanding of CNS plasticity. However, insight into the mechanisms responsible for the reduced strength of PV inhibition is limited, as hearing loss-induced changes in excitatory currents or voltage-gated properties of RS neurons, which were not assessed, could explain the apparent reduction of inhibitory gain. In addition, there is an uncertain relationship between the concept of homeostasis and the measurements of inhibitory strength which display complicated and dissimilar changes in recovered and non-recovered animals.

Essential revisions:

Correlation versus causation:

1) The main conclusion of the studies is that the initial reduction in PV interneuron-mediated inhibition predicts whether the cell will recover from the trauma. The authors provide correlative data in support of this argument, but do not show causation. The primary question is whether this effect is just one of a large number of changes in cortical circuits or is key to the recovery process. Several observations raise some concerns about this hypothesis. First, the amount of reduction in PV inhibition in the non-recovering RS cells following ouabain exposure (~35%) is similar (~40%) to that observed in the FS cells that recovered from noise-induced trauma. Second, on a cell-by-cell basis the inhibition strength in non-recovered animals does not appear to predict recovery – the data presented in Figure 3 show that there is a similar loss of inhibition in both cases and yet one cell partially recovered and one did not. Of note, the summary data for PV-mediated inhibition doesn't appear to match data illustrated in the figures (see panel 2E, day-3 and day 50, summarized in 2F, and compare to bottom panel of Figure 2). Day 0 recordings were obtained 2-14 hours after noise exposure or ouabain application. Moreover, the example rasters suggest that GABAergic inhibition was completely unchanged on Day 0, whereas the summary data show a significant, but incomplete, reduction (except for the non-recovery group). It would be more intuitive for the reader to view examples that matched the summary data. More importantly, the time course of reduced inhibitory gain after each of these hearing loss protocols is not shown with great enough temporal resolution. In addition, it is not clear why the time course of inhibitory change differed between recovered and non-recovered groups. If all changes are homeostatic, then the kinetics should be consistent across all groups. Third, the authors note that all cells within an animal showed the same behavior (e.g. all partially recover or all fail to recover). Given the intrinsic variability among RS cells with regard to PV inhibition and stochastic changes following injury, one might predict that some cells would show recovery and others would not. These findings suggest that the rapid reduction in PV inhibition and increase in firing rate may not be key to inducing compensatory changes in cortical circuits. Indeed, the dramatic shift in responsiveness to low frequency sounds after noise exposure appears to occur before there is much change in inhibition, suggesting that there are other important sites of regulation. Without some direct manipulations of PV interneurons, the role of this decrease in PV inhibition remains speculative.

2) The interpretation seems to be that GABAergic changes emerge over the first 5 days. In contrast, there is a rapid unmasking of low frequency responsiveness (Figure 2, top two panels) on Day 0. Since data were collected during a 12 hour interval on Day 0, and data were merged for the subsequent 5 days, the actual time course over which inhibitory strength declines is uncertain. By way of comparison to nerve transection experiments in NHPs, short latency (hours) changes to GABAergic transmission are observed, raising the possibility that there is a very rapid response to the manipulations that may not be revealed by the sampling resolution used in this study. Since the study seeks to characterize the temporal relationship between inhibitory homeostatic mechanisms and other measures of recovery following hearing loss, a better temporal resolution could be important to the overall interpretation. A similar question arises for the data presented in Figure 3.

3) Concerning the difference between recovered and unrecovered animals, perhaps there is a critical amount of survival that is required to promote reorganization, and a greater number of spiral ganglion cells survived in the animals that recovered. Are there within animal comparisons that could establish whether the level of damage correlates with the neurophysiological findings (e.g., recovered versus non-recovered)? Notably, the authors do not provide histological evidence that the degree of synapse loss induced by these manipulations in this strain of mice is similar to that reported by Kujawa and Lieberman (C57BL/6 versus CBA/Cj, which were not examined by Kujawa). This issue is important, as different strains of mice exhibit different sensitivities for noise induced hearing loss. Again, with reference to the NHP somatosensory literature, when animals receive amputations or dorsal rhizotomies that deafferent a whole region of sensory cortex, there are large areas that fail to reorganize. However, when a sufficient number of nerve fibers remain, then that is enough for the neuraxis to reorganize.

Mechanisms responsible for the reduced PV neuron mediated suppression of RS cell firing

1) The apparent reduction of PV-mediated inhibition (i.e., failure of a light-evoked PV cell activity to block spontaneous discharge) could be explained by an increase in tonic excitatory currents. That is, if tonic excitatory currents increased following hearing loss, then a normal level of inhibition would appear to be less efficacious at suppressing discharge. The interpretation of reduced inhibitory gain would be more convincing if there were data or analyses to confirm that excitatory gain did not change, at least during the first 5 days. However, in vitro and in vivo experiments in the context of homeostasis suggest that excitatory and voltage-gated currents do change with similar manipulations, and the increased spontaneous discharge rate observed in the current study is consistent with this idea.

2) The interpretation that inhibitory gain declines (Figure 2) is based on the suppression of responses of RS neurons and could be explained by reduced FS cell excitability, reduced GABA release, or reduced GABAA receptors located postsynaptically. Are there observations that support any of these specific mechanisms? For example, were a sufficient number of FS cell waveforms recovered through spike sorting to determine how these cells responded to blue light pulses following hearing loss? If there are multiple mechanisms, do they occur over a similar time course? What happens to FS cell spontaneous firing rates following hearing loss? Since they are recurrently connected to pyramidal cells which, themselves, display increased spontaneous rates, one might expect an increase. Does the shift of RF after hearing loss apply for PV neurons as well?

Relationship to prior studies

1) The authors make the following statement in the Introduction: "It is unclear how these distinct biomarkers of cortical reorganization following injury -receptive field remapping, spontaneous rate increases, and central gain increases – are coordinated over the days and weeks following varying degrees of sensory nerve injury. Nor is it certain whether these plasticity processes precede or follow homeostatic regulation of intracortical inhibition."

There are many relevant publications investigating how similar outcome measures are coordinated during the somatosensory reorganization that accompanies peripheral and central nerve injury. Changes to inhibition and concomitant reorganization of receptive fields are generally evident as soon as recordings can be obtained (e.g., Turnbull and Rasmusson, Somatosens Mot Res. 7:365, 1990; Calford and Tweedale, Somatosens Mot Res. 8:249, 1991; Rasmusson et al., Somatosens Mot Res. 10:69, 1993). More recently, a brief period of inactivity was demonstrated to induce a short latency reduction of IPSP amplitude in cortex (Lu et al., PNAS 111:1616, 2014). In addition, the rapid unmasking of subthreshold receptive fields is known to be kinetically distinct from the NMDA receptor-dependent reorganization that takes place in the weeks that follow sensory dennervation, especially in NHP models of peripheral dennervation. Thus, there are likely to be many mechanisms that operate across different time courses. In fact, homeostatic synaptic responses (reduced inhibitory or increased excitatory gain) to deprivation may permit "silent" inputs to become suprathreshold through NMDAR-dependent plasticity. It would be helpful to place the current findings in the context of a very well-developed literature.

2) Concerning the title. The results show plasticity that begins soon after the manipulation and lasts for a long time. The question is whether all of this plasticity can be termed "homeostatic." Homeostasis covers many specific cellular mechanisms that take place after sensory deprivation or during learning. It is important to make it clear exactly what, where, and when you are referring to homeostasis (e.g., Are all of the observations, from day 0-50, a single homeostatic process?). Since the changes seem to occur proximal to the hearing loss, this issue could be addressed by specifying what is meant by the word "compensatory" and specifying what is meant by the term "homeostatic." The Abstract does refer to homeostasis as occurring "during the first days," but it is not clear that homeostatic mechanisms emerge over that long a period of time.

Additional issues

1) The authors state that noise exposure induces permanent loss of the ABR, assessed through wave 1 amplitude; however, Figure 2 shows almost 50% recovery by 30 days. Please revise or provide further justification for this statement.

2) The authors do not provide histological evidence that the degree of synapse loss induced by these manipulations in this strain of mice is similar to that reported by Kujawa and Lieberman (C57BL/6 versus CBA/Cj, which were not examined by Kujawa). This issue is important, as different strains of mice exhibit different sensitivities for noise induced hearing loss.

3) The authors indicate that "optetrode assemblies into the 32 kHz region of the A1 tonotopic map and recorded RS units with high-frequency receptive fields, low-intensity thresholds to broadband noise stimulation and strong feedforward inhibition from PV neurons." It would be helpful if the authors indicated what proportion of sampled cells exhibited these properties. Was there a correlation between the degree of PV-mediated inhibition and the response of the neurons to peripheral injury?

---

## [Author Response]

Essential revisions:

Correlation versus causation:

1) The main conclusion of the studies is that the initial reduction in PV interneuron-mediated inhibition predicts whether the cell will recover from the trauma. The authors provide correlative data in support of this argument, but do not show causation.

It seems there was a misunderstanding here. We never intended to claim that changes in PV-mediated inhibition were necessary or sufficient(i.e., causal) for the changes in RS firing properties. The primary motivation and innovation of our study was to apply a new approach to monitor day-by-day changes in PV-mediated inhibition while tracking long-term changes in functional recovery following sensory nerve damage. Our work covers new ground by combining 1) recordings from isolated principal neurons over a period of several months, 2) with single day resolution 3) in unanesthetized animals 4) before and after 5) varying degrees of 6) selective primary afferent lesions, 7) while concurrently measuring dynamics in intracortical inhibition from 8) a genetically isolated subtype of GABA neuron. Another important, novel element of this work is that it tracks the restoration of cortical processing for spared afferent fibers in the damaged portion of the nerve. The typical approach is to completely lesion a restricted region of the retina, skin surface or basilar membrane or else entirely transect a peripheral nerve. With these methods, cortical reorganization can only be studied from the perspective of a competitive remapping of viable regions of the peripheral receptor into deafferented zones of cortex. This has been described extensively in the literature. With our approach, we can contrast this competitive invasion (e.g., low-frequency unmasking in Figure 2) against the less-studied but clinically important question of how cortical neurons recover sensitivity to intact peripheral inputs within a damaged region of the nerve.

We submitted this work in a Short Report format (bearing in mind all the associated space limitations) because we thought it leveraged a few clever contemporary technologies to make a succinct and surprising *correlative* observation about cortical network recovery from varying degrees of peripheral injury. Indeed, the major advance of this study isn’t the technological innovation described above, but rather the striking, dynamic and variable expression of cortical plasticity that it allowed us to document. Of course, we also see the value of a full-length report that delves into the underlying mechanisms. We are pursuing this line of questioning in separate studies. We regret leaving the reviewers with the impression that we were claiming a mechanism and have carefully revised the manuscript to alleviate this confusion. For example, we now write, “Importantly, this is not to say that auditory response normalization was mediated by disinhibition; our observations are correlative and do not speak to the necessity nor sufficiency of changes in PV networks for functional recovery.”

We want to be unequivocal that we would have carried out our experiments differently, had we set out to test whether changes in PV-mediated inhibition caused varying patterns of functional recovery. Adding the experiments to test causality to this revision would have essentially amounted to adding an entirely new project that, as we argue below, would not have had a high chance of yielding interpretable results.

For starters, testing the contribution of other possible mechanisms raised by the reviewers such as changes in extra-synaptic (tonic) excitation or inhibition, changes in intrinsic properties or even sourcing inhibitory changes to the PV neurons themselves (i.e., presynaptic) versus putative pyramidal neurons (i.e., postsynaptic) would be best addressed with whole cell recordings, not extracellular recordings of action potentials. Of course, this would be very challenging with our approach to using adult (16 weeks) mice with complex past histories of nerve damage and unestablished trajectories of recovery. Providing a strong test of these mechanisms by measuring spikes alone is not feasible, and we recognized this going into the study. On the other hand, there is no way to record from single neurons for 50 days with a whole cell recording (50 minutes is hard enough). This study was focused on describing long-term changes with single day (and single cell) resolution.

As a second point, proving that changes in PV-mediated inhibition was necessary and sufficient for functional recovery would have required us to block and/or artificially introduce a rapid swing in PV signaling following injury on the timescale reported here (several days). We considered these experiments, but decided that they were unlikely to yield interpretable results. Technically, conventional opsins are optimally used on time scales of tens or hundreds of milliseconds, while step opsins or DREADDS are effective on a scale of tens of minutes. But no available method can continuously activate or silence a genetically targeted cell type for days, as would be required to recreate or block the PV dynamics described here without directly engaging homeostatic mechanisms or, worse yet, drastically changing intracellular pH and all the unwanted complications that come therein. Conceptually, those experiments would be likely to fail because PV neurons are reciprocally connected with recurrently interconnected networks of principal neurons. Any approach that purported to replace or introduce PV signaling by uniformly activating or silencing PV neurons would have to contend with the heterogeneous, non-linear effects that arise from not only removing/adding inhibition from principal neurons, but also the downstream effects of changing principal neuron outputs onto other principal neurons and back onto PV neurons. It just wouldn’t work for a densely interconnected local circuit like those found in primary sensory cortex (and the literature repeatedly bears this out). Such an approach might be possible, for example, if one were testing the necessity or sufficiency of a long-range modulatory input for a plasticity in the local cortical network. But that’s a separate issue.

We did state (and continue to state) that early changes in PV-mediated inhibition are *predictive* of later changes in functional recovery, but that, in itself, does not imply that these changes are causal. Knowing the depth of PV modulation in the early days following injury allows an observer to make an accurate prediction of how/whether functional networks will eventually recover sensory processing (e.g., Figure 4).

The primary question is whether this effect is just one of a large number of changes in cortical circuits or is key to the recovery process. Several observations raise some concerns about this hypothesis. First, the amount of reduction in PV inhibition in the non-recovering RS cells following ouabain exposure (~35%) is similar (~40%) to that observed in the FS cells that recovered from noise-induced trauma.

Without belaboring the points raised above, we agree with the reviewers that changes in PV circuits cannot be directly equated with changes in functional response properties. Other mechanisms are likely to contribute. The reviewers’ critiques made it clear to us that we did not do a very good job of communicating this point in the original version.

We have made extensive changes to the text and figures to ensure that we are communicating a clear and consistent message on this point in the revised manuscript. We have replaced all of our example units to ensure that they better represent the group trends, have reanalyzed all data to include the key time intervals shortly following injury and have completely overhauled the Introduction and Discussion sections. Yes, the reviewers are absolutely right that the degree of PV-mediated inhibition, measured at an arbitrary time point, cannot be equated with the degree of nerve damage. We emphasize that it is the depth of change (i.e., the first derivative) of PV-mediated inhibition shortly following nerve damage, and not the absolute change, that distinguishes mice that recover sound processing from those that do not. With that said, the changes in PV-mediated inhibition are distinct for all three groups, as shown in Figure 4.

Second, on a cell-by-cell basis the inhibition strength in non-recovered animals does not appear to predict recovery – the data presented in Figure 3 show that there is a similar loss of inhibition in both cases and yet one cell partially recovered and one did not. Of note, the summary data for PV-mediated inhibition doesn't appear to match data illustrated in the figures (see panel 2E, day-3 and day 50, summarized in 2F, and compare to bottom panel of Figure 2). Day 0 recordings were obtained 2-14 hours after noise exposure or ouabain application. Moreover, the example rasters suggest that GABAergic inhibition was completely unchanged on Day 0, whereas the summary data show a significant, but incomplete, reduction (except for the non-recovery group). It would be more intuitive for the reader to view examples that matched the summary data.

The reviewers were correct to point out that our example units conveyed a different message than the statistical analysis of the group data. This was an error that we have remedied in the revised manuscript. In terms of the group data, our main findings remain unchanged: RS units in mice that recover auditory thresholds following bilateral ouabain show a transient loss and recovery of inhibition, whereas units from mice that do not recover show only a monotonic loss of inhibition (Figure 3 top vs. middle row). Further to this, on a cell-by-cell basis, the loss of inhibition 6-10 days after ouabain predicts whether the animal ultimately recovers or does not (Figure 4).

We have selected all new example units that better represent these findings from a noise-exposed mouse (Figure 2), a ouabain-treated mouse that recovered (Figure 3) and a ouabain-treated mouse that did not recover auditory thresholds (Figure 3). The data in panel 2F was chosen to illustrate our analysis strategy from an additional unit; those data do not come from the unit above. The figure legend has been revised to make this clear.

More importantly, the time course of reduced inhibitory gain after each of these hearing loss protocols is not shown with great enough temporal resolution.

We addressed this comment by reanalyzing all of our data and updating all statistical analyses to reflect a finer-grained analysis of changes over the first days following auditory nerve injury. All summary data in Figure 2 and Figure 3 now present changes with day-by-day resolution within the first 5 days following noise exposure (Figure 2) or ouabain/sham treatment (Figure 3). Data points from the first five days are plotted in red, to enhance the visual contrast with the preceding and following recordings. The new analyses and plots show a small (~10%) but significant decrease in inhibition at the first time point after noise exposure (Figure 2), a large, significant decrease in inhibition at the first time point after ouabain treatment in mice that recover function (Figure 3, top) and no change in inhibition at the earliest time points in ouabain-treated mice that do not recover function (Figure 3, middle) or in control mice treated with saline (Figure 3, bottom).

In addition, it is not clear why the time course of inhibitory change differed between recovered and non-recovered groups. If all changes are homeostatic, then the kinetics should be consistent across all groups.

Yes, this is one of the more intriguing observations in the study. Though we have yet to identify the factor that enables the cortex to recover auditory processing (or, alternatively, the factor that thwarts this process), we have identified predictors for the eventual mode of recovery, as illustrated in Figure 4.

As suggested by the reviewers and described below, we performed due diligence and test all plausible sources available to us in our dataset to explain the status of recovery. Nearly all of these findings are negative. For example, we cannot explain the qualitative difference in inhibitory change between the Recovered and Non-Recovered group based on the completeness of auditory nerve damage as the reduction in ABR wave 1 amplitude is massive, selective to Type-I spiral ganglion neurons and equivalent between groups. Nor can we attribute the difference to changes in outer hair cell-based cochlear amplification, because the DPOAE thresholds are at equivalent and normative levels in both groups. The data described in Figure 2–Figure 4 suggest that individual differences in the PV circuit dynamics are associated with the mode of recovery following a similar injury. As the reviewers note and we now make patently clear in the revised manuscript, these observations are correlative but they set the stage for future studies that would to optogenetically commandeer long-range neuromodulatory inputs to “steer” the mode of recovery towards one or another functional endpoint.

We addressed this point in our revised Discussion and took care not to use the term “homeostatic” when specifically referring to changes in the Non-Recovered group.

Third, the authors note that all cells within an animal showed the same behavior (e.g. all partially recover or all fail to recover). Given the intrinsic variability among RS cells with regard to PV inhibition and stochastic changes following injury, one might predict that some cells would show recovery and others would not.

We made sure to cite the work in the revised Discussion that might lead one to this expectation. Beyond that, it is not clear how we can further address this comment. The data are the data. The observation that neurons from the same mouse change in a similar way would be less likely to occur by chance than the finding that changes were variable between neurons of a given mouse. The variability in RS unit response properties before and after nerve damage is now provided as a new figure, Figure 4—figure supplement 1. There is some variance in RS response properties but it does not predict the function outcome and RS units.

These findings suggest that the rapid reduction in PV inhibition and increase in firing rate may not be key to inducing compensatory changes in cortical circuits. Indeed, the dramatic shift in responsiveness to low frequency sounds after noise exposure appears to occur before there is much change in inhibition, suggesting that there are other important sites of regulation. Without some direct manipulations of PV interneurons, the role of this decrease in PV inhibition remains speculative.

This comment overlaps with many of the points raised above. We agree that the causal role of PV inhibition is speculative and make this limitation clear in the revised Discussion. The revised analysis shows that PV-mediated inhibition is significantly reduced (albeit only by ~10%) at the first time point, when the receptive field has shifted to low frequencies.

2) The interpretation seems to be that GABAergic changes emerge over the first 5 days. In contrast, there is a rapid unmasking of low frequency responsiveness (Figure 2, top two panels) on Day 0. Since data were collected during a 12 hour interval on Day 0, and data were merged for the subsequent 5 days, the actual time course over which inhibitory strength declines is uncertain. By way of comparison to nerve transection experiments in NHPs, short latency (hours) changes to GABAergic transmission are observed, raising the possibility that there is a very rapid response to the manipulations that may not be revealed by the sampling resolution used in this study. Since the study seeks to characterize the temporal relationship between inhibitory homeostatic mechanisms and other measures of recovery following hearing loss, a better temporal resolution could be important to the overall interpretation. A similar question arises for the data presented in Figure 3.

Our previous plots did not show the finest temporal resolution possible during the early period of recovery. We now plot changes in all functional markers with single day resolution for all groups in Figure 2 and 3. The revised analysis reveals that feedforward inhibition from local PV networks is significantly reduced at the first time point in both of groups that recover sound processing. Importantly, a significant drop in PV-mediated inhibition is not found for another week in mice that fail to recover after massive nerve damage (and don’t change at all in sham-treated mice).

3) Concerning the difference between recovered and unrecovered animals, perhaps there is a critical amount of survival that is required to promote reorganization, and a greater number of spiral ganglion cells survived in the animals that recovered.

We explicitly compared cochlear afferent synapse survival with the degree of functional recovery in our prior study (Chambers et al., Neuron 2016, Figure 7D) and concluded that there was no obvious relationship. In an earlier study, we have explicitly compared ABR wave 1b amplitude to the number of surviving cochlear afferent synapses and determined that it can be used as a reliable proxy for afferent innervation of the inner hair cell both in normal and ouabain-treated ears (Yuan et al., JARO 2013, Figure 9). Here, we compare mean ABR wave 1b amplitude in ouabain-treated mice that recover versus those that do not and did not see any indication of a difference between Recovered and Non-Recovered mice (Figure 3). Because of our prior study that explicitly compared cochlear synapse survival to cortical recovery and our original work demonstrating that wave 1b could be a proxy for cochlear afferent synapse innervation, we reasoned that there was not a strong rationale to process cochleaes in ouabain-treated mice.

However, to address the reviewers’ comments as completely as possible we have reanalyzed the data linking ABR wave 1b and refer the reader to the newly added Figure 3—figure supplement 1. This revised, more granular, analysis describes markers of cochlear denervation and functional recovery of auditory threshold from each mouse tested and the lack of correspondence between the auditory nerve compound action potential amplitude (i.e., wave 1b) and recovery of function at the level of the cortex is made more apparent.

Are there within animal comparisons that could establish whether the level of damage correlates with the neurophysiological findings (e.g., recovered versus non-recovered)?

Other than the correlations provided in Figure 4, no. However, we agree that it is important to show the features that *fail* to predict recovery, and not just the factors that do show an association. The peripheral predictors are provided in Figure 3—figure supplement 1, described above. The within-neuron predictors are now provided in the newly added Figure 4—figure supplement 1.

Notably, the authors do not provide histological evidence that the degree of synapse loss induced by these manipulations in this strain of mice is similar to that reported by Kujawa and Lieberman (C57BL/6 versus CBA/Cj, which were not examined by Kujawa). This issue is important, as different strains of mice exhibit different sensitivities for noise induced hearing loss.

That’s true, we did not dissect the cochleaes from these mice and perform quantitative immunolabeling on primary afferent synapses. To be fair, there are exceedingly few studies of cortical plasticity following peripheral denervation that recover and directly analyze the sensory receptor epithelium. We go further than most by quantifying the compound action potential from the partially denervated peripheral nerve (i.e., ABR wave 1b) and establishing that the damage is neural, rather than affecting the transduction apparatus itself (i.e., by measuring otoacoustic emissions). As described above, we have performed the cochlear anatomy studies in prior work with ouabain-based denervation and this work provides a justification to use ABR wave 1b amplitude as a proxy for the survival of cochlear afferent synapses onto inner hair cells.

Regarding cochlear synaptopathy with the moderate noise exposure protocol, we naturally had many discussions with Kujawa and Liberman (my neighbors at Mass. Eye and Ear) about the cochlear synaptopathy in other mouse strains. Though the reviewers are correct in stating that many published studies on selective cochlear synaptopathy with noise exposures that cause only a temporary threshold shift (TTS) have been performed on the CBA mouse strain, TTS and cochlear synaptopathy have, in fact, been described using C57BL6 and C67BL6 hybrid strains (e.g., Wan et al., *eLife* 2014). We include a subset of the data published in that paper (Figure 5). The published work from our lab and our colleagues’ labs provide the requisite proof that ABR wave 1b amplitude can be used as a proxy for Type-I ganglion cell innervation of the inner hair cell both after ouabain-treatment or noise exposures that cause only a temporary threshold shift.

Author response image 1.Additional published data demonstrating the link between temporary ABR threshold shift, permanent wave 1 ABR amplitude loss and permanent loss of cochlear afferent synapses in a C57BL6 hybrid mouse strain.Cochlear afferent synapses are quantified by measuring GluR2+ puncta on peripheral auditory nerve processes that oppose synaptic ribbons, a presynaptic release protein complex at the base of the inner hair cell.**DOI:**
http://dx.doi.org/10.7554/eLife.21452.010

Again, with reference to the NHP somatosensory literature, when animals receive amputations or dorsal rhizotomies that deafferent a whole region of sensory cortex, there are large areas that fail to reorganize. However, when a sufficient number of nerve fibers remain, then that is enough for the neuraxis to reorganize.

This is an important point that we now address our revised Discussion. For starters, one of the main points of our paper is that cortical sound processing recovers more completely, with less drastic short-term compensatory changes, after moderate auditory nerve injury (noise exposure) than for extreme nerve injury. This is consistent with the larger literature on cortical plasticity following deafferentation mentioned by the reviewers.

As a second point, as described above, there is no evidence in our dataset (namely, ABR wave1b amplitude and DPOAE threshold) or in our previous study (Chambers et al., Neuron 2016) that the degree of peripheral injury can explain how well ouabain-treated mice recover auditory processing or not. This isn’t to say that a peripheral factor cannot explain the mode of recovery, just that we looked at the most obvious markers and nothing fits the pattern of cortical recovery.

Thirdly, it is important to point out that dorsal rhizotomy is a complete deafferentation, whereas both modes of injury employed here spare a subset of afferent synapses within the zone of injury. This is a key difference that distinguishes our model of auditory denervation from focal retinal lesions or nerve transection in primate models of somatosensory and visual cortex plasticity.

Finally, while the reviewers are correct to point out that the degree of injury can explain the completeness of ‘filling in’ from spared regions of the skin surface, other studies in NHP and rodent models, also make the point that factors unrelated to the degree of injury, such as the manner of use of skin surface can also exert a profound influence on adult somatotopic maps (Clark et al., 1988 Nature, Allard et al., 1991 J. Neurophys, Xerri et al., 1996 J Neurophys; Merzenich & Jenkins 1993 J Hand Therapy; Wang et al., 1995 Nature; Xerri et al., 1998, J Physiol., Polley et al., 1999, Neuron). Therefore, it is not unreasonable to surmise that activity-dependent features unrelated to the number of spared peripheral inputs could bias, if not explain outright, whether mice recovery auditory thresholds or not. We refer the reviewers to the revised Discussion where we address this issue more completely.

Mechanisms responsible for the reduced PV neuron mediated suppression of RS cell firing

*1) The apparent reduction of PV-mediated inhibition (i.e., failure of a light-evoked PV cell activity to block spontaneous discharge) could be explained by an increase in tonic excitatory currents. That is, if tonic excitatory currents increased following hearing loss, then a normal level of inhibition would appear to be less efficacious at suppressing discharge. The interpretation of reduced inhibitory gain would be more convincing if there were data or analyses to confirm that excitatory gain did not change, at least during the first 5 days. However,* in vitro *and* in vivo *experiments in the context of homeostasis suggest that excitatory and voltage-gated currents do change with similar manipulations, and the increased spontaneous discharge rate observed in the current study is consistent with this idea.*

We agree. There are good reasons to question whether voltage-gated currents, excitatory synaptic currents and/or extrasynaptic currents might also be contributing to the functional recovery described here. Every approach has its strength and limitations. Ours is the first approach to track changes in receptive fields, central gain, spontaneous firing rate and PV-mediated inhibition in the same neuron over a period of months surrounding a controlled injury. This approach has many merits but we could not measure EPSCs or tonic currents from the neurons we have isolated and therefore cannot identify their contribution one way or the other. On the other hand, we can monitor the strength of PV-mediated inhibition over time and relate these dynamics to the progressive recovery of function. This is something that no previous study has been able to do, not at the level of an individual neuron at least, and this proves to be a useful measurement that might have been missed with acute measurements that offer more penetration insight into possible mechanisms.

As described above, we have reworked our Discussion paragraphs to focus on possible mechanisms and caveats in the interpretation of this work. We now cite work related to each of the mechanisms raised by the reviewers. We make it clear that changes in PV circuits described here are likely not the sole mechanism. Indeed, enhanced sensitivity to glutamatergic inputs may work synergistically with dynamic changes in PV input to bring about enhanced responsiveness following peripheral denervation. As an example, we refer the reviewers to preliminary fluorescent in situ hybridization data from an ongoing study in our lab that provides single cell labeling of mRNA for an AMPA and GABAAreceptor subunit (Figure 6). One can appreciate that compared to a sham-treated age-matched control mouse, the cell in the ouabain-treated mouse has *less message for the GABAA receptor and more message for the AMPA receptor*.

Author response image 2.Triple fluorescence in situ hybridization of an example neuron from the auditory cortex of a ouabain-treated adult mouse or an age-matched sham-treated control.Transcripts for Gria2, which encodes an AMPA receptor subunit, and GABRA1, which encodes a GABAA receptor subunit, are visualized in the peri-nuclear compartment. Compared to the sham control, glutamate receptor transcripts are increased and GABAA receptor transcripts are decreased. These data are part of a separate study that examines the interplay between glutamate sensitization and GABA disinhibition after auditory nerve damage.**DOI:**
http://dx.doi.org/10.7554/eLife.21452.011

2) The interpretation that inhibitory gain declines (Figure 2) is based on the suppression of responses of RS neurons and could be explained by reduced FS cell excitability, reduced GABA release, or reduced GABAA receptors located postsynaptically. Are there observations that support any of these specific mechanisms? For example, were a sufficient number of FS cell waveforms recovered through spike sorting to determine how these cells responded to blue light pulses following hearing loss? If there are multiple mechanisms, do they occur over a similar time course? What happens to FS cell spontaneous firing rates following hearing loss? Since they are recurrently connected to pyramidal cells which, themselves, display increased spontaneous rates, one might expect an increase. Does the shift of RF after hearing loss apply for PV neurons as well?

All good questions. We sought to address some of these points by quantifying laser-evoked, sound-evoked and spontaneous spiking in fast-spiking, PV-expressing interneurons. We provide our analysis in the newly added Figure 1—figure supplement 1 where we show, unfortunately, that it was essentially impossible to hold a FS neuron for two or more consecutive days. Not only were these waveforms less commonly encountered but, presumably because of their comparatively smaller size, the recordings were far less stable than RS units. We concluded that we just didn’t have the basis to perform the complementary analysis on FS units without a robust sample of stable chronic recordings. However, we revised the Discussion to touch on these important points. We made this limitation clear in the revised Introduction and by including the new figure.

Relationship to prior studies

1) The authors make the following statement in the Introduction: "It is unclear how these distinct biomarkers of cortical reorganization following injury -receptive field remapping, spontaneous rate increases, and central gain increases – are coordinated over the days and weeks following varying degrees of sensory nerve injury. Nor is it certain whether these plasticity processes precede or follow homeostatic regulation of intracortical inhibition."

There are many relevant publications investigating how similar outcome measures are coordinated during the somatosensory reorganization that accompanies peripheral and central nerve injury. Changes to inhibition and concomitant reorganization of receptive fields are generally evident as soon as recordings can be obtained (e.g., Turnbull and Rasmusson, Somatosens Mot Res. 7:365, 1990; Calford and Tweedale, Somatosens Mot Res. 8:249, 1991; Rasmusson et al., Somatosens Mot Res. 10:69, 1993). More recently, a brief period of inactivity was demonstrated to induce a short latency reduction of IPSP amplitude in cortex (Lu et al., PNAS 111:1616, 2014). In addition, the rapid unmasking of subthreshold receptive fields is known to be kinetically distinct from the NMDA receptor-dependent reorganization that takes place in the weeks that follow sensory dennervation, especially in NHP models of peripheral dennervation. Thus, there are likely to be many mechanisms that operate across different time courses. In fact, homeostatic synaptic responses (reduced inhibitory or increased excitatory gain) to deprivation may permit "silent" inputs to become suprathreshold through NMDAR-dependent plasticity. It would be helpful to place the current findings in the context of a very well-developed literature.

We cited several seminal studies relating to somatosensory deafferentation in NHP in our original manuscript and cite additional work, including some of those mentioned above, in the revised work. The revised manuscript features an overhauled Introduction and Discussion with many new references that hopefully do a better job of putting our findings into context, as the reviewers have suggested. Space is limited in the Short Report format and the literature on cochlear denervation and auditory system plasticity is also well-developed and important to cite in this context. We have done our best to cite the seminal and most relevant work from the auditory, somatosensory and visual literature.

At the same time, it is also important to note the differences between the earlier studies and the findings described here. Most work in the existing literature was performed with acute measurements of unit activity in anesthetized animals at one or two times post-injury, as compared to separate control animals. If measurements of intracortical inhibition were performed, they were almost invariably at single time points and are not specific to any subtype of GABA neuron.

2) Concerning the title. The results show plasticity that begins soon after the manipulation and lasts for a long time. The question is whether all of this plasticity can be termed "homeostatic." Homeostasis covers many specific cellular mechanisms that take place after sensory deprivation or during learning. It is important to make it clear exactly what, where, and when you are referring to homeostasis (e.g., Are all of the observations, from day 0-50, a single homeostatic process?). Since the changes seem to occur proximal to the hearing loss, this issue could be addressed by specifying what is meant by the word "compensatory" and specifying what is meant by the term "homeostatic." The Abstract does refer to homeostasis as occurring "during the first days," but it is not clear that homeostatic mechanisms emerge over that long a period of time.

We revised the title so that it no longer includes the term “homeostatic” because it is unclear whether/how some of longer term changes we describe (or lack of changes in the Non-Recover group) reflect homeostatic mechanisms.

Additional issues

1) The authors state that noise exposure induces permanent loss of the ABR, assessed through wave 1 amplitude; however, Figure 2 shows almost 50% recovery by 30 days. Please revise or provide further justification for this statement.

The change in ABR amplitude during the early period of temporary threshold shift (e.g., at the day 2 measurement time in Figure 2) reflects and immediate and permanent loss of cochlear synapses, but an additional transient changes to stereocilia tip links or cochlear supporting cells that affect mechanoelectric transduction. With TTS, the changes in the organ of Corti reverse and thresholds go back to normal and part of the ABR amplitude loss is recovered. The persistent decrease in ABR amplitude *when unaccompanied by a difference in DPOAE or ABR threshold* arises from the cochlear afferent synaptopathy. We revised the manuscript to make this clear in the sixth paragraph of the Results. The same pattern is observed in row one of Author responses Figure 1, taken from Wan et al., *eLife* 2014.

2) The authors do not provide histological evidence that the degree of synapse loss induced by these manipulations in this strain of mice is similar to that reported by Kujawa and Lieberman (C57BL/6 versus CBA/Cj, which were not examined by Kujawa). This issue is important, as different strains of mice exhibit different sensitivities for noise induced hearing loss.

The issue has already been raised and addressed in Point #3 of the Essential revisions, above.

3) The authors indicate that "optetrode assemblies into the 32 kHz region of the A1 tonotopic map and recorded RS units with high-frequency receptive fields, low-intensity thresholds to broadband noise stimulation and strong feedforward inhibition from PV neurons." It would be helpful if the authors indicated what proportion of sampled cells exhibited these properties. Was there a correlation between the degree of PV-mediated inhibition and the response of the neurons to peripheral injury?

The requested analyses are now provided in the top left panel of Figure 4—figure supplement 1. We show that most single units, particularly for the noise exposure group for which the above claim was made, have best frequencies between 16-32 kHz (Figure 4—figure supplement 1). We further show that the PV-mediated inhibition and response thresholds for broadband noise bursts before damage was > 85% and less than 40 dB SPL, respectively, was fairly uniform across our sample (Figure 4—figure supplement 1), and were not correlated with each other (r = 0.04, Figure 4—figure supplement 1). Finally, we show that neither the strength of inhibition prior to auditory nerve damage, nor auditory response thresholds prior to nerve damage could predict the eventual recovery of each marker (r = 0.05 and 0.06, respectively, Figure 4—figure supplement 1). Beyond the specific requests made here, we mined our dataset for features that predict the eventual mode of recovery. The best candidates are described in Figure 4.